# Antimicrobial Activity of Several Cineole-Rich Western Australian *Eucalyptus* Essential Oils

**DOI:** 10.3390/microorganisms6040122

**Published:** 2018-12-03

**Authors:** Fahad S. Aldoghaim, Gavin R. Flematti, Katherine A. Hammer

**Affiliations:** 1Infection Control Department, Prince Sultan Military Medical City, Riyadh 12233, Saudi Arabia; dr.fahd123@gmail.com; 2School of Molecular Sciences, The University of Western Australia, Crawley, WA 6009, Australia; gavin.flematti@uwa.edu.au; 3School of Biomedical Science, The University of Western Australia, Crawley, WA, 6009, Australia

**Keywords:** minimum inhibitory concentration, monoterpenes, volatile oil, oil mallee, 1,8-cineole, eucalyptol

## Abstract

Essential oils from the Western Australian (WA) *Eucalyptus* mallee species *Eucalyptus loxophleba*, *Eucalyptus polybractea*, and *Eucalyptus kochii* subsp. *plenissima* and subsp. *borealis* were hydrodistilled from the leaves and then analysed by gas chromatography–mass spectrometry in addition to a commercial *Eucalyptus globulus* oil and 1,8-cineole. The main component of all oils was 1,8-cineole at 97.32% for *E. kochii* subsp. *borealis*, 96.55% for *E. kochii* subsp. *plenissima*, 82.95% for *E. polybractea*, 78.78% for *E. loxophleba* 2, 77.02% for *E. globulus*, and 66.93% for *E. loxophleba* 1. The *Eucalyptus* oils exhibited variable antimicrobial activity determined by broth microdilution, with *E. globulus* and *E. polybractea* oils showing the highest activities. The majority of microorganisms were inhibited or killed at concentrations ranging from 0.25% to 8.0% (*v*/*v*). *Enterococcus faecalis* and *Candida albicans* were the least susceptible organisms, whilst *Acinetobacter baumannii* was the most sensitive. In conclusion, all oils from WA *Eucalyptus* species showed microorganism inhibitory activity, although this varied according to both the *Eucalyptus* species and the microorganism tested. These data demonstrate that WA *Eucalyptus* oils show activity against a range of medically important pathogens and therefore have potential as antimicrobial agents.

## 1. Introduction

*Eucalyptus* is a genus of plants native to Australia and some islands to the north of Australia. It comprises over 700 species, most of which are endemic to Australia [1,2]. Since *Eucalyptus* species are able to grow under a variety of climatic and edaphic conditions, they have been extensively introduced in areas outside Australia, including the United States, the Middle East, India, and South America [2]. *Eucalyptus* oil is obtained from the leaves by steam distillation and contains predominantly volatile terpenes and aromatic compounds, the most abundant typically being the monoterpenoid 1,8-cineole [3,4]. British and European pharmacopoeias specify that *Eucalyptus* oil must contain at least 70% 1,8-cineole when the oil is used for medicinal purposes [5].

Eucalypt plants have been used in traditional medicine in Australia for thousands of years. The Australian Aborigines use the leaves for medicinal purposes to treat a range of ailments including wounds and fungal infections [6]. The leaf extracts, including the essential oil, are currently widely used in perfumery and cosmetic products and to a lesser extent as a therapeutic agent. The current medicinal use is based on the range of biological effects exhibited by the oils in vitro, including antioxidant [7], anti-inflammatory, analgesic [8], and antimicrobial activities [9,10,11,12,13]. Clinical trials with *Eucalyptus* oil and the major component 1,8-cineole (eucalyptol) have been performed to evaluate their efficacy in the treatment of a diverse range of conditions and diseases, including respiratory disorders [14,15], oral hygiene [16], and head lice infestation [17].

Although a number of researchers have previously investigated the antimicrobial activities of *Eucalyptus* essential oils, relatively little is known about the composition and activity of several Western Australian (WA) *Eucalyptus* oils. The aim of this work was therefore to determine the chemical composition of the essential oil of WA *Eucalyptus* species, namely, *Eucalyptus loxophleba*, *Eucalyptus polybractea*, *Eucalyptus kochii* subsp. *plenissima*, and *E. kochii* subsp. *borealis*, also known as “oil mallees”, and to examine their antimicrobial activities against a range of common pathogenic bacteria.

## 2. Materials and Methods

### 2.1. Plant Material

Fresh leaves of *E. loxophleba* 1, *E. polybractea* (subspecies not identified) (grown in Armadale, Western Australia), *E. loxophleba* 2, *E. kochii* subsp. *plenissima*, and *E. kochii* subsp. borealis (grown in Kalannie in the Wheatbelt Region of Western Australia) were harvested in March 2015 (about 700 g each) and immediately transported to the School of Molecular Sciences at The University of Western Australia (UWA), Crawley, Western Australia. Commercial *Eucalyptus* oil from *Eucalyptus globulus* (Thursday Plantation, Australia) and 1,8-cineole (99.0% purity; Fluka Chemika) were used for comparison.

### 2.2. Extraction of Essential Oils

A portion of ca. 150 g of leaf material was cut into small pieces and added to 400 mL of de-ionised water in a blender (Waring, HGB2WTS3, New Hartford, CT, USA). The material was macerated for 1 min and then combined with two additional portions (ca. 450 g total) and subjected to hydrodistillation in a Clevenger-type apparatus for approximately 3 h. The oil/water emulsion produced was collected and stored at 4 °C overnight to separate the essential oil from the residual water. The essential oil was then removed and stored in an amber glass bottle at room temperature until further use. 

### 2.3. Gas Chromatography–Mass Spectrometry (GC–MS) Analysis of Essential Oils

GC–MS was performed with a Shimadzu GCMS-QP2010 (Kyoto, Japan). GC columns used included a Rtx-5 column (5% diphenyl-dimethyl-polysiloxane, 30 m × 0.25 mm × 0.1 μm film thickness, Restek, Bellefonte, PA, USA), and a DB-wax column (polyethylene glycol, 30 m × 0.25 mm × 0.25 μm film thickness, J&W Scientific, Folsom, CA, USA). Helium was used as the carrier gas with a constant flow rate of 1.0 mL/min on both columns. A scan range of *m*/*z* 45–400 and a solvent delay of 5 min were used with splitless injections of 1.0 µL for 1 min. The ion source was set to 230 °C, and the transfer line temperature to 250 °C. The oven temperature program was 40 °C, held for 1 min then ramped at 7 °C/min to 250 °C and held for 10 min. Retention indices (RI) were calculated on both columns using the same linear gradient method with comparison to an n-hydrocarbon mixture (Sigma-Aldrich, St Louis, MO, USA, p/n 46827-U). The main peaks in the total ion chromatogram of each oil were then integrated using the MS software, and the relative percentage abundance of peaks was determined.

### 2.4. Microorganisms

Microorganisms included a range of Gram-positive and Gram-negative bacteria and a yeast. Gram-positive bacteria were *Enterococcus faecalis* ATCC 29212, *Enterococcus faecalis* (vancomycin-resistant enterococci VRE) ATCC 51299, *Staphylococcus aureus* ATCC 29213, methicillin-resistant *S. aureus* (MRSA) NCTC 10442, and *Staphylococcus epidermidis* NCTC 11047. Gram-negative bacteria were *Escherichia coli* ATCC 25922, *Salmonella enterica* subsp. *enterica* serovar Typhimurium ATCC 13311, *Acinetobacter baumannii* NCTC 7844, and *Pseudomonas aeruginosa* ATCC 27853. *Candida albicans* ATCC 90028 was included as a representative yeast. The strains were cultured on blood agar at 35 °C, for 24 h for the bacteria and for 48 h for *C. albicans* prior to use in antimicrobial testing.

### 2.5. Evaluation of Antimicrobial Activity

*Eucalyptus* oils and 1,8-cineole were initially screened for activity against one Gram-positive strain (*S. aureus* ATCC 29213) and one Gram-negative reference strain (*E. coli* ATCC 25922) using an agar diffusion assay. The inocula were prepared by inoculating the organisms onto the blood agar and incubating overnight at 35 °C. the colonies were then suspended in 0.85% saline, and the suspension was adjusted to a turbidity of 0.5 McFarland (10^8^ colony forming units (CFU)/mL) using a nephelometer. The suspension was then swab-inoculated onto Mueller–Hinton agar, and 8 mm diameter wells were punched into the agar. Volumes of 25 µL and 50 µL of each *Eucalyptus* oil and 1,8-cineole were then aliquoted into the wells. Trimethoprim (5 µg/disc) was used as a positive control. After incubation at 35 °C for 24 h, zones of inhibition were measured, and the results were reported in millimetres (mm). Each oil was tested at least three times on separate occasions, and mean values were calculated.

The minimum inhibitory concentration (MIC) of each oil was determined using a broth microdilution method based on protocols published by the Clinical and Laboratory Standard Institute (CLSI) [18,19]. The method was modified slightly by incorporating a final concentration of 0.001% Tween 80 to enhance oil solubility. In brief, each *Eucalyptus* oil was serially diluted two-fold in 100 µL volumes in a 96-well microtitre tray so that after inoculation with 100 µL of inoculum per well, the final concentrations ranged from 8.0% to 0.0016% (*v*/*v*). Using the known density of 1,8-cineole (0.9267 g/mL at 20 °C) [20] as a conversion factor, these percentage values corresponded to a range of 74.136 mg/mL–0.145 mg/mL of *Eucalyptus* oil. A positive growth control well containing growth medium and 0.001% Tween 80 but without *Eucalyptus* oil was included. The inocula were prepared from overnight cultures as described above and adjusted to 0.5 McFarland for bacteria, which corresponded to approximately 10^8^ CFU/mL, or 1.0 McFarland for *C. albicans*, which corresponded to approximately 10^7^ CFU/mL [18,19]. The suspensions were diluted as required to result in final inocula concentrations of approximately 5 × 10^5^ CFU/mL. After inoculation and incubation for 24 h for bacteria and for 48 h for *C. albicans*, the MIC was determined visually as the lowest concentration of the *Eucalyptus* oil preventing microbial growth.

The minimum bactericidal concentration (MBC) or minimum fungicidal concentration (MFC) was determined by removing 10 µL volumes from each well showing no visible growth and spot-inoculating onto Mueller–Hinton agar. After incubation, colonies were counted and the MBC/MFC was identified as the lowest concentration of oil that killed ≥99.9% of the inoculum. The assay was conducted three times on separate occasions, and modal MIC/MBC/MFC values were selected.

### 2.6. Statistical Analysis

Geometric means of MICs and MBC/MFCs were determined for each *Eucalyptus* oil and for each microorganism. Geometric means were also determined to examine the difference in sensitivity between Gram-positive and Gram-negative bacteria. To enable the analyses, values >8.0% were converted to the next highest doubling dilution value of 16.0%. A one-way analysis of variance (ANOVA) was used to compare the MIC results between the two *S. aureus* strains and the two *E. faecalis* strains. A *p*-value of <0.05 was considered significant.

## 3. Results

A total of 21 distinguishable compounds were detected across the leaf oil samples by GC–MS (Table 1; Figure 1). The most abundant compound was 1,8 cineole, ranging from the lowest value of 66.93% for *E. loxophleba* 1 to 97.32% for *E. kochii* subsp. *borealis*. Other compounds detected in proportions greater than 5.0% were limonene (7.52%), *p*-cymene (5.53%), and γ-terpinene (5.34%) in *E. globulus* oil, and 4-methyl-2-pentyl acetate in both *E. loxophleba* 1 (9.86%) and *E. loxophleba* 2 (5.53%) oils.

When the oils were screened for activity using the semi-quantitative agar diffusion assay, all *Eucalyptus* oils and 1,8-cineole produced zones of inhibition against the two test bacteria (Table 2). The largest zone of inhibition was observed for 50 µL of *E. polybractea* oil against *S. aureus* ATCC 29213. For the remaining oils, zone sizes were relatively modest and ranged from 11.0 to 16.7 mm. On the basis of this data, more comprehensive antibacterial studies were conducted, using an expanded range of test organisms.

MIC and MBC/MFC results are shown in Table 3. The *Eucalyptus* oils showed variable antimicrobial activity against the different test organisms. The MIC geometric means for test organisms ranged from 1.2% for *A. baumannii* to 14.5% for *E. faecalis* and the MBC⁄MFC geometric means ranged from 1.6% for *A. baumannii* to >8.0% for *E. faecalis* ATCC 29212 and *S. epidermidis*. The Gram-negative organism *A. baumannii* was the most sensitive to the *Eucalyptus* oils, followed by *S. enterica* Typhimurium and *E. coli*. The Gram-positive *E. faecalis* ATCC 29212 was the least susceptible. All the examined *Eucalyptus* oils, with the exception of *E. kochii* subsp. *plenissima*, showed high activity against *E. faecalis* VRE, with MIC values ranging from 2.0% to 8.0% *v*/*v*. Comparison of the two strains of *S. aureus* showed that the MRSA strain was significantly more susceptible to *Eucalyptus* oils than the antibiotic-sensitive *S. aureus* strain (*p* < 0.05). Similarly, MICs obtained for *E. faecalis* VRE were also significantly different from those obtained for the susceptible *E. faecalis* strain (*p* = 0.00002).

The MIC values also showed that the different *Eucalytpus* oils tested varied in antimicrobial activity. Comparison of the geometric mean of the MICs for each oil using the results for all test organisms showed that *E. polybractea* and *E. globulus* oils displayed the highest activity with geometric means of 4.3% for both oils, followed by *E. loxophleba* 2 and *E. kochii* subsp. *borealis* (4.6%), *E. kochii* subsp. *plenissima* (5.6%), 1,8 cineole (7.0%), and *E. loxophleba* 1 (7.5%) (Table 3). *E. polybractea* oil was the only oil that inhibited the growth of all organisms at ≤8.0% (*v*/*v*). *E. globulus* oil inhibited 8/10 organisms at ≤8.0% (*v*/*v*), the exceptions being *E. faecalis* and *C. albicans*. In contrast, 1,8 cineole inhibited only 4/10 test organisms. The MICs of *E. loxophleba* 1 and *E. loxophleba* 2 oils varied substantially from each other. There was a one- to two-fold decrease in MICs with *E. loxophleba* 2 oil compared to *E. loxophleba* 1 oil for both *S. aureus* strains, *S. epidermidis*, *C. albicans*, *S. enterica* Typhimurium, and *A. baumannii*, and a one- to two-fold increase for *E. faecalis* VRE and *P. aeruginosa*. *E. kochii* subsp. *plenissima* oil showed strong antibacterial activity against *S. aureus*, *S. enterica* Typhimurium, *E. coli*, and *A. baumannii* at ≤8.0% (*v*/*v*), yet, had no activity against the two strains of *E. faecalis*, *S. epidermidis*, and *P. aeruginosa*. For *E. kochii* subsp. *borealis* oil, the most sensitive strain was *A. baumannii* with an MIC of 1.0% (*v*/*v*), followed by *S. aureus* and *E. coli* with MICs of 2.0% (*v*/*v*); the least susceptible strains were *E. faecalis* and *P. aeruginosa* (>8.0%).

Overall, for each organism and oil combination, the MIC values were often identical to, or differed by only one concentration from the MBC or MFC values for that oil (Table 3). This indicates that most oils had activity that was bactericidal or fungicidal in nature. The exceptions were the Gram-positive bacteria *S. aureus* MRSA, *E. faecalis* VRE, and *S. epidermidis*, for which the values differed by more than two concentrations.

## 4. Discussion

The major compound present in all oils was 1,8-cineole, which is in keeping with previous studies indicating that 1,8-cineole is often the major component of *Eucalyptus* oils [1,5]. The percentages of 1,8-cineole in *E. polybractea*, *E. loxophleba*, and *E. kochii* subsp. *plenissima* oils were largely in agreement with those previously reported [23,24,25]. *E. kochii* subsp. *borealis* was previously known as both *Eucalyptus oleosa* var. *borealis* and *Eucalyptus horistes* [26], and the levels of 1,8-cineole reported for the oils from these species were 75.5% [27] and 90.17% [23], respectively. The two *E. loxophleba* oils varied in composition, possibly due to differences in the local climatic and environmental conditions under which the plants grew (given that they were from different regions), as well as the genetic characteristics and the age of the trees [28]. It is also possible that they were different subspecies, as *E. loxophleba* has three subspecies, including *loxophleba*, *lissophloia*, and *gratiae*. Bignell et al. (1997) found 1,8-cineole levels of 25.2% in *E. loxophleba* subsp. *loxophleba* and 63.0% in *E. loxophleba* subsp. *lissophloia*, indicating large differences in composition [24]. Regardless, relatively little information is published on the chemical composition of WA *Eucalyptus* oils, and, as such, a comparison of the current data with previous findings is limited.

With regard to antimicrobial activity, numerous publications have described the activity of *Eucalyptus* oils against *S. aureus* and *E. coli* using the agar diffusion method [11,12]. In the current study, low to modest activity (with the exception of *E. polybractea*) was observed by agar diffusion, which concurs with these previous studies. However, although *E. polybractea* oil was the only oil that exhibited a considerable zone of inhibition against *S. aureus*, Gilles et al. (2010) reported considerable zones of inhibition against *S. aureus*, ranging from 25.4 to >90 mm, for oils from four other *Eucalyptus* species [13]. These differences in results may be attributable to variation in the *Eucalyptus* oils tested, as well as to variations in the experimental conditions. Finally, there was little correlation between the zone of inhibition and the MIC results, particularly for *E. polybractea* oil which showed the largest zone of inhibition against *S. aureus* but not the lowest MICs. This could be due to the presence of components in the *E. polybractea* oil that are relatively more water-soluble and able to diffuse further through the agar. The lack of correlation between the results from the two assays suggests that the two methods are not necessarily comparable. The agar diffusion method has long been regarded as problematic for the antimicrobial testing of natural products [29]. Because of issues with inconsistent diffusion of antimicrobial components through the agar, potential evaporation of volatile components, and lack of standardization between laboratories [30], data generated by agar diffusion must be regarded as largely qualitative, and the assay as useful for screening purposes only.

When more quantitative testing was conducted using the broth microdilution method, all eucalyptus oils showed antimicrobial activity. This is consistent with previous studies, verifying that both eucalyptus oil and 1,8-cineole have activity against a wide range of Gram-positive and Gram-negative bacteria and yeasts [11,12]. The current study found considerable variation in activity between the different eucalyptus oils, which could be a reflection of differences in their chemical composition. Comparison of the activity of *Eucalyptus* oils to 1,8-cineole alone confirmed the importance of the other moderate and minor components in *Eucalyptus* oils in relation to the MIC and MBC/MFC values obtained. Comparison of geometric mean MICs for all oils showed that *E. globulus* and *E. polybractea* oils had the lowest values, with a geometric mean of 4.3% (*w*/*v*), and *E. loxophleba* 1 and 1,8-cineole had the highest values, with geometric means of 7.5% and 7% (*w*/*v*), respectively. These results align with those reported by Cimanga et al. (2002). They showed that oils from *Eucalyptus deglupta*, *Eucalyptus saligna*, *Eucalyptus urophylla*, and *Eucalyptus propinque*, which all contain relatively high percentages of 1,8 cineole (>30%), exhibited similar or lower antimicrobial activity than oils from *Eucalyptus alba*, *Eucalyptus robusta*, *Eucalyptus citriodora*, and *Eucalyptus tereticornis*, which contain low percentages of 1,8-cineole (<10%) [31].

It was not surprising that the two *E. loxophleba* oils had substantially different antimicrobial activity, given that these two oils varied in composition and that *E. loxophleba* comprises a complex of several sub-species, rather than a single species [32]. Whilst it is not known which sub-species of *E. loxophleba* the oils in the current study were from, the variability in activity may be attributed to differences in the composition such as the presence of trans-pinocarveol (4.66%) in *E. loxophleba* 2 oil, which has been shown to have broad-spectrum antimicrobial activity [32]. These observations also suggest that a high 1,8-cineole content is not necessarily fully responsible for the activity of *Eucalyptus* oil and that moderate and minor compounds also play a vital role in the overall activity. Components other than 1,8-cineole may contribute to the activity as a result of their individual actions, by acting in combination with each other, or by acting in combination with 1,8-cineole. Such interactions between 1,8-cineole and other essential oil components have been demonstrated in previous studies [32,33].

It was unexpected that, according to the MIC/MBC values, the Gram-negative bacteria were, broadly speaking, more susceptible to *Eucalyptus* oils than the Gram-positive bacteria. This susceptibility of Gram-negative bacteria may be due to the presence of certain oil components, such as *p*-cymene, terpinolene, 1,8-cineole, and cis-geraniol, which can cause the discharge of Gram-negative outer membrane lipopolysaccharide and increase the permeability of the cytoplasmic membrane [34]. Of the Gram-negative bacteria, *A. baumannii* was the most sensitive to the *Eucalyptus* oils. This result is particularly relevant, given that *Acinetobacter* spp. are emerging globally as problematic antibiotic-resistant pathogens [35]. Our results are in agreement with several previous studies that also found *Acinetobacter* spp. to be relatively more sensitive than other Gram-negative species [22,36]. Very few researchers have investigated the potential mechanisms underlying this increased sensitivity; however, it could be due to differences in the outer membrane composition or efflux pumps of *Acinetobacter* species and *Enterobacteriaceae* [37,38]. Finally, many previous reports on the antimicrobial activity of *Eucalyptus* oil are difficult to compare with the current study because of differences in the methods used to assess the antimicrobial activity and differences in the chemical composition of the oils.

A number of clinical studies indicate that *Eucalyptus* oil and 1,8-cineole have significant potential as therapeutic agents, due to several different properties. For example, the benefits of 1,8-cineole for airway disease [39] and *Eucalyptus* oil as an insect repellent [40] have recently been reviewed. The toxicity of *Eucalyptus* oil has been extensively studied both in vitro and in animal studies, with the latter indicating that 1,8-cineole has low toxicity [40]. Furthermore, studies with human volunteers indicate that, when applied correctly, the oil has relatively low allergenicity and toxicity [40]. Whilst many studies have investigated the antimicrobial activity of *Eucalyptus* oil in vitro, fewer clinical studies have been performed. As such, there is no clear overall picture of the clinical usefulness of *Eucalyptus* oil as an antimicrobial agent. Further studies are required to determine how the antimicrobial properties of *Eucalyptus* oil can be best therapeutically utilised.

## 5. Conclusions

This study showed that *Eucalyptus* oils from some selected WA species had moderate antimicrobial activity, which varied according to the Eucalypt species and the test microorganism. The data suggest that WA *Eucalyptus* oils are potentially a good source of antimicrobial agents, particularly against Gram-negative bacteria. As such, further studies with additional test organisms and additional oil samples are warranted.

## Figures and Tables

**Figure 1 microorganisms-06-00122-f001:**
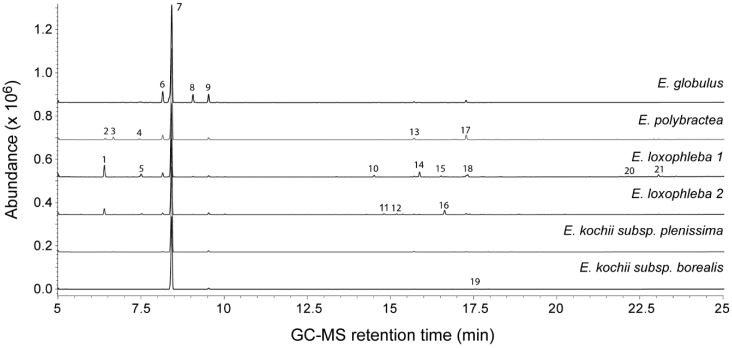
Overlaid GC–MS total ion chromatograms (TIC) of the *Eucalyptus* oils used in this study. Each number correlates with specific compounds identified in Table 1.

**Table 1 microorganisms-06-00122-t001:** Relative percentage of *Eucalyptus* oil components present at concentrations greater than 0.1%, determined using GC–MS.

Compound	RI (DB-wax)	RI (Rtx-5)	*Eucalyptus globulus*	*Eucalyptus polybractea*	*Eucalyptus loxophleba* 1	*E. loxophleba* 2	*Eucalyptus kochii* Subsp. *plenissima plenissima*	*E. kochii* subsp. *borealis*
4-Methyl-2-pentyl acetate (1)	1109 (1110 ^a^)	–	–	–	9.86	5.53	–	–
*β*–Pinene (2) *	1110 (1116 ^b^)	974 (981 ^b^)	–	1.07	–	–	–	–
Sabinene (3)	1123 (1123 ^b^)	–	–	1.98	–	–	–	–
*β*-Myrcene (4) *	1164 (1160 ^c^)	992 (992 ^b^)	–	0.59	0.57	–	–	–
*α*-Phellandrene (5) *	1166 (1166 ^b^)	1003 (1007 ^b^)	0.43	–	2.75	–	–	–
Limonene (6) *	1201 (1201 ^b^)	–	7.52	3.67	3.52	1.58	0.53	–
1,8-Cineole (7) *	1214 (1213 ^b^)	1032 (1030 ^b^)	77.02	82.95	66.93	78.78	96.55	97.32
γ-Terpinene (8) *	1248 (1238 ^b^)	1058 (1074 ^b^)	5.34	–	0.37	–	–	–
*p*-Cymene (9) *	1272 (1261 ^b^)	1026 (1027 ^b^)	5.53	1.50	1.11	1.77	1.39	1.34
*α*-Gurjunene (10)	1541 (1536 ^d^)	1409 (1412 ^e^)	–	–	1.48	–	–	–
3-Pinanone (11)	1558 (1576 ^a^)	1173 (1163 ^a^)	–	–	–	0.89	–	–
Pinocarvone (12) *	1578 (1565 ^a^)	1162 (1160 ^a^)	–	–	–	0.79	–	–
Terpinen-4-ol (13) *	1609 (1618 ^d^)	1177 (1176 ^e^)	0.56	1.39	0.52	0.39	0.61	0.12
Aromadendrene (14) *	1619 (1625 ^d^)	1440 (1446 ^e^)	–	–	4.37	0.3	–	–
*allo*-Aromadendrene (15) *	1659 (1667 ^d^)	1462 (1466 ^e^)	–	–	0.94	–	–	–
*trans*-Pinocarveol (16)	1664 (1675 ^d^)	1137 (1127 ^f^)	–	–	–	4.66	–	–
*α*-Terpineol (17) *	1702 (1709 ^a^)	1190 (1189 ^a^)	1.49	3.67	1.45	1.22	0.33	0.11
Ledene (18) *	1708 (1706 ^d^)	1498 (1504 ^e^)	–	–	1.87	–	–	__
Verbenone (19)	1722 (1728 ^d^)	1210 (1228 ^a^)	–	–	–	–	–	0.36
epi-Globulol (20)	2025 (2039 ^d^)	1561 (1566 ^e^)	–	–	0.44	–	–	–
Globulol (21)	2091 (2103 ^d^)	1585 (1595 ^e^)	–	–	1.90	0.3	–	–
Total identified compounds			97.89	96.82	98.08	96.21	99.41	99.25

Notes: – not detected. * confirmed with commercial standard. Compound numbers correlate with Figure 1. Retention indices (RI) values in parenthesis indicate literature values from ^a^ P.J. Linstrom and W.G. Mallard, Eds., NIST Chemistry WebBook, NIST Standard Reference Database Number 69, National Institute of Standards and Technology, Gaithersburg MD, 20899, http://webbook.nist.gov, (retrieved 28 June 2016). ^b^ Flavornet at http://www.flavornet.org (retrieved 28 June 2016). ^c^ LRI and Odour database at http://www.odour.org.uk, (retrieved 28 June 2016). ^d^ Reference [21]. ^e^ Reference [22]. ^f^ Reference [1].

**Table 2 microorganisms-06-00122-t002:** Zones of bacterial growth inhibition (mean and standard deviation in mm) resulting from agar diffusion of two different volumes of *Eucalyptus* oils.

*Eucalyptus* Oils	*Staphylococcus aureus* ATCC 29213		*Escherichia coli* ATCC 25922	
	25 µL	50 µL	25 µL	50 µL
*E. globulus*	13.0 ± 1.0	15.3 ± 0.6	11.3 ± 0.6	15.0 ± 0.0
*E. loxophleba* 1	15.3 ± 0.6	16.7 ± 0.6	14.7 ± 0.6	16.7 ± 0.6
*E. loxophleba* 2	13.0 ± 0.6	15.0 ± 0.0	12.3 ± 0.6	15.3 ± 0.6
*E. polybractea*	28.0 ± 0.0	29.5 ± 0.7	14.0 ± 1.0	16.7 ± 0.6
*E. kochii* subsp. *plenissima*	13.0 ± 0.0	15.7 ± 0.6	13.0 ± 0.0	13.0 ± 0.0
*E. kochii* subsp. *borealis*	12.7 ± 0.6	14.7 ± 0.6	11.3 ± 0.6	13.0 ± 0.0
1,8 Cineole	11.0 ± 0.0	12.7 ± 0.6	13.3 ± 0.6	14.3 ± 0.6
Trimethoprim 5 µg	27.7 ± 0.6		26.3 ± 0.6	

**Table 3 microorganisms-06-00122-t003:** Susceptibility of microorganisms to *Eucalyptus* oils (MIC % *v*/*v*) determined by the broth microdilution assay.

Essential Oil	Parameter ^a^	*S. aureus* ATCC 29213	*S. aureus* MRSA NCTC 10442	*Enterococcus faecalis* ATCC 29212	*E. faecalis* VRE ATCC 51299	*Staphylococcus epidermidis* NCTC 11047	*Candida albicans* ATCC 90028	*Salmonella* Typhimurium ATCC 13311	*E. coli* ATCC 25922	*Pseudomonas aeruginosa* ATCC 27853	*Acinetobacter baumannii* NCTC 7844	Geometric Mean of the MIC
*E. globulus*	MIC	4	2	>8	4	4	>8	0.5	8	8	2	4.3
	MBC/MFC	4	>8	>8	>8	>8	>8	0.5	8	8	2	
*E. loxophleba* 1	MIC	>8	8	>8	2	>8	>8	8	8	4	2	7.5
	MBC/MFC	>8	>8	>8	8	>8	>8	8	8	4	4	
*E. loxophleba* 2	MIC	4	4	>8	4	8	8	2	8	>8	0.25	4.6
	MBC/MFC	>8	4	>8	4	>8	>8	4	8	>8	0.25	
*E. polybractea*	MIC	8	4	8	2	2	8	8	4	4	2	4.3
	MBC/MFC	>8	8	8	>8	>8	>8	8	4	8	2	
*E. kochii* subsp.	MIC	2	4	>8	>8	>8	8	2	2	>8	2	5.6
*plenissima*	MBC/MFC	8	4	>8	>8	>8	>8	4	2	>8	4	
*E. kochii* subsp.	MIC	2	4	>8	4	8	8	4	2	>8	1	4.6
*borealis*	MBC/MFC	4	8	>8	4	>8	>8	4	2	>8	2	
1,8 Cineole	MIC	>8	>8	>8	8	>8	>8	2	1	>8	1	7.0
	MBC/MFC	>8	>8	>8	>8	>8	>8	4	4	>8	1	
Geometric mean	MIC	5.4	4.9	14.5	4.4	8.0	10.8	2.7	3.6	9.7	1.2	
	MBC/MFC	9.7	8.8	14.5	9.7	>8.0	>8.0	3.6	4.4	9.8	1.6	

^a^ MIC, minimum inhibitory concentration; MBC, minimum bactericidal concentration; MFC, minimum fungicidal concentration. Values are expressed as % (*v*/*v*).

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
