# Peer review of "Antimicrobial Activity of Several Cineole-Rich Western Australian Eucalyptus Essential Oils"

_microorganisms, 2018, doi:10.3390/microorganisms6040122_

Round 1

Reviewer 1 Report

This is an interesting research, well written and structured. However, some criticisms occurred, the first is the lack of some important technical information (see specific comments). In addition, I found the lack of a possible explanation or a comparison of the observed antimicrobial activity against A. baumanni with other similar researches existing in literature. I strongly suggest to improve this part of the discussion an to update some references about Eucalyptus oils (i.e. Barbosa LC et al., “Chemical Variability and Biological Activities of Eucalyptus spp. Essential Oils.” Molecules. 2016; Sebei K et al., “Chemical composition and antibacterial activities of seven Eucalyptus species essential oils leaves. Biol Res. 2015)

Specific comments:

In the title the name “Eucalyptusmust be written in italic (as appears in the manuscript). I think that is a typewritten error.

Material and Methods

Page 2

Lines 40-45: “Gram-positive bacteria were Enterococcus faecalis ATCC 29212, Enterococcus faecalis (vancomycin-resistant enterococci VRE) ATCC 51299, Staphylococcus aureus ATCC 29213, methicillin-resistant Staphylococcus aureus (MRSA) NCTC 10442 and Staphylococcus epidermidis NCTC 11047. Gram-negative bacteria were Escherichia coli ATCC 25922, Salmonella enterica subsp. enterica serovar Typhimurium ATCC 13311, Acinetobacter baumannii NCTC 7844 and Pseudomonas aeruginosa ATCC 44 27853.” I suggest something as: “Enterococcus faecalis ATCC 29212, Enterococcus faecalis ATCC 51299 (vancomycin-resistant enterococci, VRE), Staphylococcus aureus ATCC 29213, methicillin-resistant Staphylococcus aureus NCTC 10442 (MRSA) and Staphylococcus epidermidis NCTC 11047 were included as gram-positive bacteria, while Escherichia coli ATCC 25922, Salmonella enterica subsp. enterica serovar Typhimurium ATCC 13311, Acinetobacter baumannii NCTC 7844 and Pseudomonas aeruginosa ATCC 44 27853 as gram-negative one. Candida albicans ATCC 90028 was also included as yeast.”

Page 3

Line 2: the authors used the term “screened” referring to the agar-well diffusion test; it’s correct but I suggest to explain that this test was performed only against one-gram positive and one-gram negative representative strain.

Line 4: “Inocula were prepared by suspending growth from an overnight culture into 0.85% saline and adjusting to approximately 108 CFU/mL.”

Please specific

1. the volume and the type of the medium used for overnight culture as well as the temperature;

2. how the density of approximately 108 CFU/mL was determined.

Line 6: add a space in “25μl”

Lines 7: delete “The antibiotic”

Line 8: I think that is more correct “35 ± 2°C” here and in all the manuscript.

Lines 14-15: the authors stated “after inoculation, final concentration ranged from 8.0% to 0.00016% (v/v)” I understand the sense, but I think that this part can be better explained adding, for example, the quantity of the bacterial inoculation. Please check.

Lines 21-22: see comments in line 4, point 2.

Results

Page 5

Lines 6: I don’t like the term “see” and I suggest to replace with “observed”

Line 7: delete “small to” and left only “modest” or replace with “negligible diameter of growth inhibition” Line 18: “All Eucalyptus oils with the exception of E. kochii subsp. plenissima were highly active” replace with “All the examined Eucalyptus oils, with the exception of E. kochii subsp. Plenissima, resulted highly active”

Line 21: delete “the susceptible”

Page 6

Line 5: “MIC data also showed that the different Eucalytpus oils varied in antimicrobial activity” I suggest something such as: “MIC values showed the diversified antimicrobial activity of the tested Eucalytpus oils”

Line 10: “The oil from E. polybractea was the only oil that inhibited the growth of all organisms at ≤ 8.0% (v/v).” This sentence has not a well sound. I suggest, here and in all the text, to avoid writing “the oil from” and to use “E. polybractea oil” and so on.

Lines 11-12: I prefer the use of a percentage rather than “eight of the 10 organisms” or “four of the 10 test”. At limit, it can be written as 8/10 and 4/10.

Discussion

Page 7

Line 10: “obtained previously” I suggest to replace with “reported in literature”

Line 23: read comment related to the term “see”

Page 8

Lines 15-26: the most interesting result of this research is the high activity against A. baumannii and, for this, I think that this part must be discussed in more in-depth way. The mentioned difference in outer membrane is referred to an old research and cannot be considered so important. I suggest to read and add as references:

Knezevic P et al. Antimicrobial activity of Eucalyptus camaldulensis essential oils and their interactions with conventional antimicrobial agents against multi-drug resistant Acinetobacter baumannii. J Ethnopharmacol. (2016)

Mulyaningsih et al. Antibacterial activity of essential oils from Eucalyptus and of selected components against multidrug-resistant bacterial pathogens. Pharm Biol. (2011)

Author Response

Reviewer 1 comments

This is an interesting research, well written and structured. However, some criticisms occurred, the first is the lack of some important technical information (see specific comments). In addition, I found the lack of a possible explanation or a comparison of the observed antimicrobial activity against A. baumanni with other similar researches existing in literature. I strongly suggest to improve this part of the discussion and to update some references about Eucalyptus oils (i.e. Barbosa LC et al., “Chemical Variability and Biological Activities of Eucalyptus spp. Essential Oils.” Molecules. 2016; Sebei K et al., “Chemical composition and antibacterial activities of seven Eucalyptus species essential oils leaves. Biol Res. 2015)

RESPONSE: Thank you for the feedback. Please see our responses to these individual comments below

Specific comments:

In the title the name “Eucalyptus” must be written in italic (as appears in the manuscript). I think that is a typewritten error.

RESPONSE: This has been italicised.

Material and Methods

Page 2

Lines 40-45: “Gram-positive bacteria were Enterococcus faecalis ATCC 29212, Enterococcus faecalis(vancomycin-resistant enterococci VRE) ATCC 51299, Staphylococcus aureus ATCC 29213, methicillin-resistant Staphylococcus aureus (MRSA) NCTC 10442 and Staphylococcus epidermidis NCTC 11047. Gram-negative bacteria were Escherichia coli ATCC 25922, Salmonella enterica subsp. enterica serovar Typhimurium ATCC 13311, Acinetobacter baumannii NCTC 7844 and Pseudomonas aeruginosa ATCC 44 27853.”

I suggest something as: “Enterococcus faecalis ATCC 29212, Enterococcus faecalis ATCC 51299 (vancomycin-resistant enterococci, VRE), Staphylococcus aureus ATCC 29213, methicillin-resistantStaphylococcus aureus NCTC 10442 (MRSA) and Staphylococcus epidermidis NCTC 11047 were included as gram-positive bacteria, while Escherichia coli ATCC 25922, Salmonella enterica subsp. enterica serovar Typhimurium ATCC 13311, Acinetobacter baumannii NCTC 7844 and Pseudomonas aeruginosa ATCC 44 27853 as gram-negative one. Candida albicans ATCC 90028 was also included as yeast.”

RESPONSE: Thank you for this suggestion. The wording has been revised.

Page 3

Line 2: the authors used the term “screened” referring to the agar-well diffusion test; it’s correct but I suggest to explain that this test was performed only against one-gram positive and one-gram negative representative strain.

RESPONSE: This suggestion has been incorporated into the manuscript.

Line 4: “Inocula were prepared by suspending growth from an overnight culture into 0.85% saline and adjusting to approximately 108 CFU/mL.”

Please specific

1. the volume and the type of the medium used for overnight culture as well as the temperature;

2. how the density of approximately 108 CFU/mL was determined.

RESPONSE: These details have been added to the manuscript.

Line 6: add a space in “25μl”

RESPONSE: This has been done.

Lines 7: delete “The antibiotic”

RESPONSE: This has been done.

Line 8: I think that is more correct “35 ± 2°C” here and in all the manuscript.

RESPONSE: Incubation temperatures are routinely reported and published as a single value without a range. With this in mind, and to keep the manuscript concise, we have elected not to change the way temperatures are displayed in the manuscript.

Lines 14-15: the authors stated “after inoculation, final concentration ranged from 8.0% to 0.00016% (v/v)” I understand the sense, but I think that this part can be better explained adding, for example, the quantity of the bacterial inoculation. Please check.

RESPONSE: The volumes for the serial dilutions and for the inocula have been added to the text. 

Lines 21-22: see comments in line 4, point 2.

RESPONSE: Further explanation has been provided.

Results

Page 5

Lines 6: I don’t like the term “see” and I suggest to replace with “observed”

RESPONSE: This change has been made.

Line 7: delete “small to” and left only “modest” or replace with “negligible diameter of growth inhibition”

RESPONSE: The text has been modified.

Line 18: “All Eucalyptus oils with the exception of E. kochii subsp. plenissima were highly active” replace with “All the examined Eucalyptus oils, with the exception of E. kochii subsp. Plenissima, resulted highly active”

RESPONSE: The text has been modified.

Line 21: delete “the susceptible”

RESPONSE: This text has been altered to clarify the meaning. This wording was in place to distinguish between the two different S. aureus strain – one of which was antibiotic sensitive and one of which was antibiotic resistant.

Page 6

Line 5: “MIC data also showed that the different Eucalytpus oils varied in antimicrobial activity” I suggest something such as: “MIC values showed the diversified antimicrobial activity of the tested Eucalytpus oils”

RESPONSE: The text has been modified.

Line 10: “The oil from E. polybractea was the only oil that inhibited the growth of all organisms at ≤ 8.0% (v/v).” This sentence has not a well sound. I suggest, here and in all the text, to avoid writing “the oil from” and to use “E. polybractea oil” and so on.

RESPONSE: The text has been modified.

Lines 11-12: I prefer the use of a percentage rather than “eight of the 10 organisms” or “four of the 10 test”. At limit, it can be written as 8/10 and 4/10.

RESPONSE: The text has been modified.

Discussion

Page 7

Line 10: “obtained previously” I suggest to replace with “reported in literature”

RESPONSE: The text has been modified.

Line 23: read comment related to the term “see”

RESPONSE: The term “seen” has been replaced with “observed”

Page 8

Lines 15-26: the most interesting result of this research is the high activity against A. baumannii and, for this, I think that this part must be discussed in more in-depth way. The mentioned difference in outer membrane is referred to an old research and cannot be considered so important. I suggest to read and add as references:

Knezevic P et al. Antimicrobial activity of Eucalyptus camaldulensis essential oils and their interactions with conventional antimicrobial agents against multi-drug resistant Acinetobacter baumannii. J Ethnopharmacol. (2016)

Mulyaningsih et al. Antibacterial activity of essential oils from Eucalyptus and of selected components against multidrug-resistant bacterial pathogens. Pharm Biol. (2011)

RESPONSE: This section of the discussion has been modified and expanded. 

It is correct that the difference in outer membrane composition is “old research”, published in 1976, but the findings are still relevant to the manuscript. Also, we could not locate any more recent papers that have specifically investigated this difference, or have attempted to explain why Acinetobacter may be comparatively more susceptible to essential oils than other Gram negative bacteria.   

Thank you for the suggested additional references. The publication by Mulyaningsih (2011) was already cited in this section of text and the publication by Knezevic et al (2016) has been added in addition so several more papers.

Reviewer 2 Report

Its a nice study to explore antimicrobial activity of the eucalytpus spp.

Table 2, please provide in the title with Mean±SE or SD, also consider it in the text.

Discussion part is too long, if possible shorten it. or add two or three lines in the begining of dicussion, explain, what you find different than the previous studies.

kindly italicize technical names in the references.

Author Response

Reviewer 2 comments

Its a nice study to explore antimicrobial activity of the eucalytpus spp. 

Table 2, please provide in the title with Mean±SE or SD, also consider it in the text.

RESPONSE: The title has been modified to include these details.

Discussion part is too long, if possible shorten it. or add two or three lines in the begining of dicussion, explain, what you find different than the previous studies. 

RESPONSE: Thank you for this comment. The authors feel that it is not possible to summarise the differences between this and previous studies in only a few lines so have elected not to add this. We have however revised the discussion and shortened it where possible. 

kindly italicize technical names in the references.

RESPONSE: This change has been made.